# Streamlined inactivation, amplification, and Cas13-based detection of SARS-CoV-2

Jon Arizti-Sanz [1,2,16], Catherine A. Freije [1,3,16], Alexandra C. Stanton[1,3], Brittany A. Petros [1,2,4], Chloe K. Boehm [1], Sameed Siddiqui[1,5], Bennett M. Shaw [1,6], Gordon Adams[1], Tinna-Solveig F. Kosoko-Thoroddsen[1], Molly E. Kemball[1], Jessica N. Uwanibe[7,8], Fehintola V. Ajogbasile[7,8], Philomena E. Eromon[7], Robin Gross[9], Loni Wronka[10], Katie Caviness[10], Lisa E. Hensley[9], Nicholas H. Bergman[10], Bronwyn L. MacInnis[1,11], Christian T. Happi[7,8,11], Jacob E. Lemieux[1,6], Pardis C. Sabeti[1,11,12,13,14,17] & Cameron Myhrvold [1,12,14,15,17 ✉]

The COVID-19 pandemic has highlighted that new diagnostic technologies are essential for controlling disease transmission. Here, we develop SHINE (Streamlined Highlighting of Infections to Navigate Epidemics), a sensitive and specific diagnostic tool that can detect SARS-CoV-2 RNA from unextracted samples. We identify the optimal conditions to allow RPA-based amplification and Cas13-based detection to occur in a single step, simplifying assay preparation and reducing run-time. We improve HUDSON to rapidly inactivate viruses in nasopharyngeal swabs and saliva in 10 min. SHINE's results can be visualized with an in-tube fluorescent readout — reducing contamination risk as amplification reaction tubes remain sealed — and interpreted by a companion smartphone application. We validate SHINE on 50 nasopharyngeal patient samples, demonstrating 90% sensitivity and 100% specificity compared to RT-qPCR with a sample-to-answer time of 50 min. SHINE has the potential to be used outside of hospitals and clinical laboratories, greatly enhancing diagnostic capabilities.

[1] Broad Institute of Massachusetts Institute of Technology (MIT) and Harvard, Cambridge, MA 02142, USA. [2] Harvard-MIT Program in Health Sciences and Technology, 77 Massachusetts Avenue, Cambridge, MA 02139, USA. [3] Program in Virology, Harvard Medical School, Boston, MA 02115, USA. [4] Harvard/ MIT MD-PhD Program, Boston, MA 02139, USA. [5] Computational and Systems Biology PhD Program, MIT, Cambridge, MA 02139, USA. [6] Division of Infectious Diseases, Department of Medicine, Massachusetts General Hospital, 55 Fruit Street Gray 730, Boston, MA 02114, USA. [7] African Centre of Excellence for Genomics of Infectious Diseases (ACEGID), Redeemer's University, Ede, Osun State, Nigeria. [8] Department of Biological Sciences, College of Natural Sciences, Redeemer's University, Ede, Osun State, Nigeria. [9] Integrated Research Facility, Division of Clinical Research, National Institute of Allergy and Infectious Diseases, National Institutes of Health, Frederick, MD 21702, USA. [10] National Biodefense Analysis and Countermeasures Center, Fort Detrick, MD 21702, USA. [11] Harvard T.H. Chan School of Public Health, 677 Huntington Avenue, Boston, MA 02115, USA. [12] Department of Organismic and Evolutionary Biology, Harvard University, 26 Oxford Street, Cambridge, MA 02138, USA. [13] Howard Hughes Medical Institute, Chevy Chase, MD 20815, USA. [14] Massachusetts Consortium on Pathogen Readiness, Boston, MA, USA. [15] Present address: Department of Molecular Biology, Princeton University, Princeton, NJ 08544, USA. [16] These authors contributed equally: Jon Arizti-Sanz, Catherine A. Freije. [17] These authors jointly supervised this work: Pardis C. Sabeti, Cameron Myhrvold. ✉email: cmyhrvol@broadinstitute.org

Point-of-care diagnostic testing is essential to prevent and effectively respond to infectious disease outbreaks. Insufficient nucleic acid diagnostic testing infrastructure[1] and the prevalence of asymptomatic transmission[2,3] have accelerated the global spread of severe acute respiratory syndrome coronavirus 2 (SARS-CoV-2)[4–6], with confirmed case counts surpassing 23 million as of August 23, 2020[7]. Ubiquitous nucleic acid testing—whether in doctor's offices, pharmacies, or pop-up testing sites—would increase diagnostic access and is essential for safely reopening businesses, schools, and country borders. Easy-to-use, scalable diagnostics with a quick turnaround time and limited equipment requirements would fulfill this major need and have the potential to alter the trajectory of this pandemic.

The current paradigm for nucleic acid diagnostic testing predominantly relies on patient samples being sent to centralized diagnostic laboratories for processing and analysis. Reverse transcriptase quantitative polymerase chain reaction (RT-qPCR), the current gold standard for SARS-CoV-2 diagnosis[8], is highly specific and sensitive but requires laboratory infrastructure for nucleic acid extraction, thermal cycling, and analysis of assay results. The need for thermocyclers can be eliminated through the use of isothermal (i.e., single temperature) amplification methods, such as loop-mediated isothermal amplification (LAMP) and recombinase polymerase amplification (RPA)[9–14]. However, for isothermal approaches to disrupt this paradigm, they would need to be low-cost, scalable, and sensitive. Colorimetric LAMP assays enable high-throughput testing with minimal equipment requirements but often require purified samples to achieve high sensitivity. Improvements to RPA enable its use with unextracted samples and with increased sensitivity but are only compatible with lateral flow-based visual readouts, which can be tedious for larger sample numbers[15]. Although Abbott's ID NOW COVID-19 test using isothermal amplification with unextracted samples can report results in 5–13 min, this technology requires expensive equipment and has low throughput (~100 samples per machine per 24-h day)[16,17]. Therefore, isothermal amplification methods still require technological advances for testing to be performed outside of laboratories at low cost and with high throughput.

Recently developed CRISPR-based diagnostics have the potential to transform infectious disease diagnosis. Both CRISPR-Cas13- and Cas12-based assays have been developed for SARS-CoV-2 detection using extracted nucleic acids as input[18–23]. One such CRISPR-based diagnostic, SHERLOCK (Specific High-sensitivity Enzymatic Reporter unLOCKing), involves two separate steps, starting with extracted nucleic acids: (1) isothermal RPA and (2) T7 transcription and Cas13-mediated collateral cleavage of a single-stranded RNA reporter[24]. Cas13-based detection is highly programmable and specific, as it relies on complementary base pairing between the target RNA and the CRISPR RNA (crRNA) sequence[24,25]. Current SHERLOCK-based diagnostics are compatible with HUDSON (Heating Unextracted Diagnostic Samples to Obliterate Nucleases), which uses heat and chemical reduction to inactivate nucleases and lyse viral particles[26]. This method eliminates the need for nucleic acid extraction but requires 30 min of incubation and has yet to be tested and validated with SARS-CoV-2 assays and with nasopharyngeal (NP) swabs. Together, these methods reduce the equipment needs and laboratory infrastructure for viral detection to solely a heating element. However, their scalability and widespread implementation is currently limited by the need for amplified products to be transferred between tubes—increasing risk of contamination and user error—and by result interpretation, which has only been automated for lateral flow-based readouts[27].

To address the current limitations of nucleic acid diagnostics, we develop SHINE (Streamlined Highlighting of Infections to Navigate Epidemics) for extraction-free, rapid, and sensitive detection of SARS-CoV-2 RNA. We establish a SHERLOCK-based SARS-CoV-2

assay[19] where amplification and Cas13-based detection are combined into a single step, decreasing user manipulations and assay time (Fig. 1a). We demonstrate that SHINE can detect SARS-CoV-2 RNA in HUDSON-treated patient samples with both a paper-based colorimetric readout and an in-tube fluorescent readout. Moreover, the fluorescent readout's results can be interpreted in an automated fashion via a newly developed pipeline within a companion smartphone application.

## Results

**Design and testing of a two-step SARS-CoV-2 SHERLOCK assay.** We first developed a two-step SHERLOCK assay that sensitively detected SARS-CoV-2 RNA at 10 copies per microliter (cp/μL). Using ADAPT, a computational design tool for nucleic acid diagnostics, we identified primers and a crRNA within open reading frame 1a (ORF1a) of SARS-CoV-2 that comprehensively captures known sequence diversity, with high predicted Cas13 targeting activity and SARS-CoV-2 specificity (Fig. 1b)[19]. Using both colorimetric and fluorescent readouts, we detected 10 cp/μL of synthetic RNA after incubating samples for ≤1 h. However, preparing the separate amplification and Cas13 detection reaction mixtures and combining each reaction mixture with each sample tested required at least 45 min for a small number (<10) of samples (Fig. 1c, d and Supplementary Fig. 1a). We tested this assay on HUDSON-treated SARS-CoV-2 viral seedstocks, detecting down to $1.31 \times 10^5$ plaque-forming units per milliliter (PFU/mL) via colorimetric readout (Supplementary Fig. 1b). Finally, we compared our two-step SHERLOCK to RT-qPCR using extracted viral RNA, demonstrating similar limits of detection using fluorescent and lateral flow-based readouts in two laboratories on different continents (Supplementary Fig. 1c, d).

**Development of a single-step SHERLOCK assay.** We sought to develop an integrated, streamlined assay that was significantly less time- and labor-intensive than the two-step SHERLOCK. However, when we combined RT-RPA (step 1), T7 transcription, and Cas13-based detection (step 2) into a single step (i.e., single-step SHERLOCK), the sensitivity of the assay decreased dramatically. This decrease was specific for RNA input and likely due to incompatibility of enzymatic reactions with the given conditions (limit of detection (LOD) $10^6$ cp/μL; Fig. 1d and Supplementary Fig. 2a). As a result, we evaluated whether additional reaction components and alternative reaction conditions could increase the sensitivity and speed of the assay. Addition of RNase H, in the presence of reverse transcriptase, improved the sensitivity of Cas13-based detection of RNA 10-fold (LOD $10^5$ cp/μL; Fig. 2a and Supplementary Fig. 2b, c). RNase H likely enhanced the sensitivity by increasing the efficiency of RT through degradation of DNA:RNA hybrid intermediates[15].

Given that each enzyme involved has optimal activity at distinct reaction conditions, we evaluated the role of different pHs, monovalent salt, magnesium, and primer concentrations on assay sensitivity. Optimized buffer, magnesium, and primer conditions resulted in an LOD of 1000 cp/μL (Fig. 2b, c and Supplementary Fig. 2d, e). We then improved the speed of Cas13 cleavage and RT to reduce the sample-to-answer time. Given the uracil-cleavage preference of Cas13a[25,28,29], detection of RNA in the single-step SHERLOCK assay reached half-maximal fluorescence in ~67% of the time when RNaseAlert was substituted for a polyU reporter (Fig. 2d, left and Supplementary Fig. 3). In addition, reactions containing SuperScript IV reverse transcriptase reached half-maximal fluorescence two times faster than RevertAid reverse transcriptase (Fig. 2d, right).

Together, these improvements resulted in an optimized single-step SHERLOCK assay that could specifically detect SARS-CoV-2

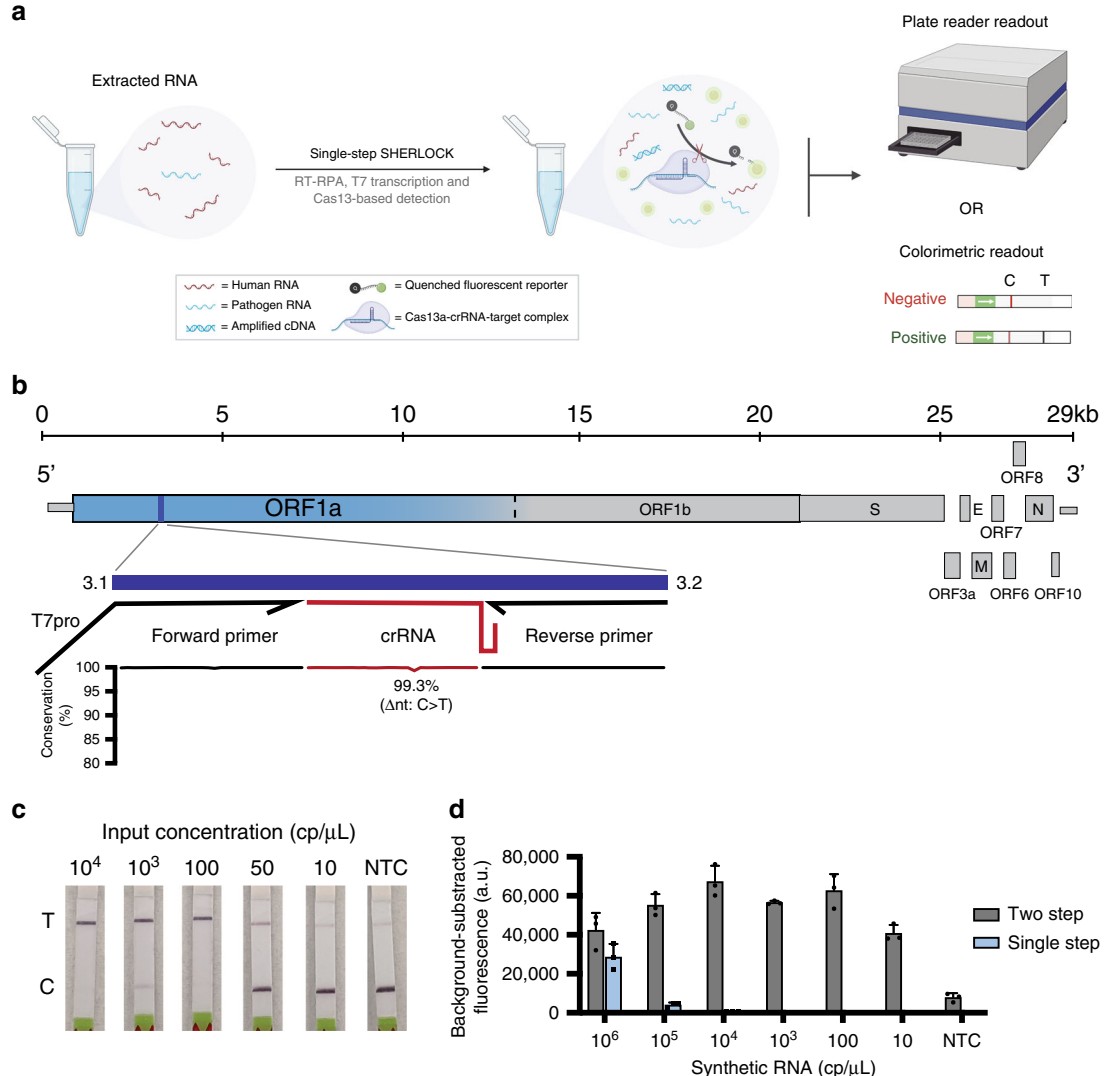

**Fig. 1 Initial assay development for SHERLOCK-based SARS-CoV-2 detection. a** Schematic of single-step SHERLOCK assays using extracted RNA with a fluorescent or colorimetric readout. RT-RPA reverse transcriptase-recombinase polymerase amplification, C control line, T test line. **b** Schematic of the SARS-CoV-2 genome and SHERLOCK assay location. Sequence conservation across the primer and crRNA-binding sites for publicly available SARS-CoV-2 genomes (see "Methods" for details). Text denotes nucleotide position with lowest percent conservation across the assay location. ORF open reading frame, T7pro T7 polymerase promoter; narrow rectangles, untranslated regions. **c** Colorimetric detection of synthetic RNA using two-step SHERLOCK after 30 min. NTC_r non-template control introduced in RPA, NTC_d non-template control introduced in detection, T test line, C control line. **d** Background-subtracted fluorescence of the two-step and original single-step SHERLOCK protocols using synthetic SARS-CoV-2 RNA after 3 h. The 1-h timepoint from this experiment is shown in Fig. 2e. NTC non-template control introduced in RPA. Center = mean and error bars = s.d. for 3 technical replicates. For **b**–**d**, source data are provided as a Source data file.

RNA with reduced sample-to-answer time and comparable sensitivity relative to our two-step assay. We tested the specificity and quantified the LOD of our optimized single-step SHERLOCK assay on synthetic SARS-CoV-2 and other human coronavirus RNA targets. Our assay detected as few as 10 cp/μL with 100% specificity using a fluorescent readout—100,000 times more sensitive than the initial assay—and 100 cp/μL using the lateral flow-based colorimetric readout (Fig. 2e, f and Supplementary Figs. 4 and 5).

We then evaluated our assay's performance on SARS-CoV-2 RNA extracted from patient NP samples. We compared our fluorescent single-step SHERLOCK assay to a previously performed RT-qPCR diagnostic using a pilot set of nine samples. We detected SARS-CoV-2 from 5 of the 5 SARS-CoV-2-positive patient samples tested, demonstrating 100% concordance with RT-qPCR, with no false positives for 4 SARS-CoV-2-negative

extracted samples or 2 non-template controls (Fig. 2g, h and Supplementary Table 1).

**App-enabled detection of SARS-CoV-2 using SHINE.** To simplify sample processing, assay output, and data interpretation, we created SHINE, a SHERLOCK-based diagnostic platform for extraction-free viral RNA detection with results interpreted by a companion smartphone application (Fig. 3a). In order to eliminate the need for purified nucleic acids and to reduce total run time, we sought to improve HUDSON[26] and test its compatibility with COVID-19 collection matrices. During optimization, we assessed HUDSON's ability to inactivate RNases by adding RNaseAlert to samples following treatment, with higher fluorescence corresponding to decreased nuclease inactivation. Through the addition of RNase inhibitors, we reduced the incubation time

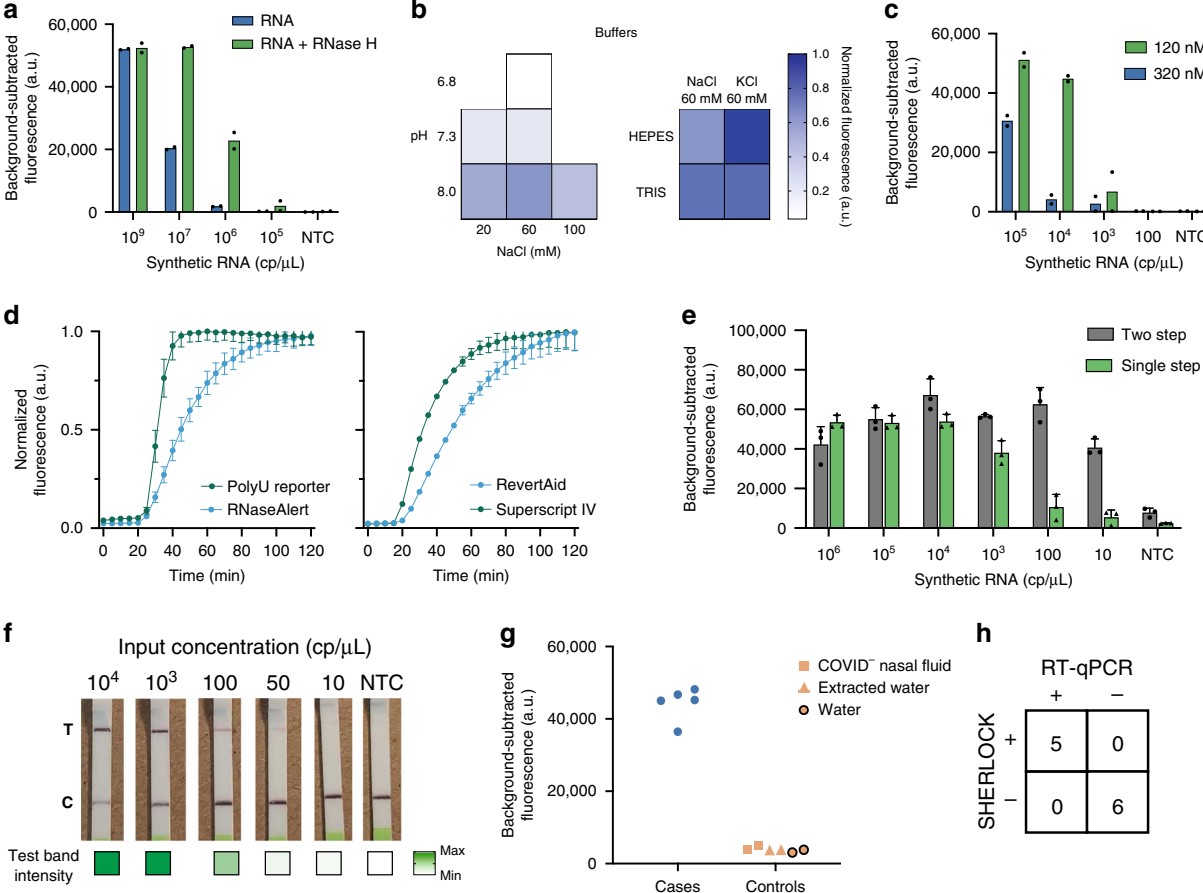

**Fig. 2 Optimization of the single-step SHERLOCK reaction. a** Background-subtracted fluorescence of Cas13-based detection with synthetic RNA, reverse transcriptase, and RPA primers (but no RPA enzymes) after 3 h. **b** Single-step SHERLOCK normalized fluorescence using various buffering conditions after 3 h. **c** Background-subtracted fluorescence of single-step SHERLOCK with synthetic RNA and variable RPA forward and reverse primer concentrations after 3 h. **d** Single-step SHERLOCK normalized fluorescence over time using two different fluorescent reporters (left) and two different reverse transcriptases (right). **e** Background-subtracted fluorescence of the original single-step and optimized single-step SHERLOCK with synthetic RNA after 1 h. Data from the 3-h timepoint from this experiment are shown in Fig. 1d. **f** Colorimetric detection of synthetic RNA input using optimized single-step SHERLOCK after 3 h. Max maximum test band intensity, 5698.4 a.u., Min minimum test band intensity, 104.4 a.u. **g** Optimized single-step SHERLOCK background-subtracted fluorescence using RNA extracted from patient samples after 1 h. **h** Concordance between SHERLOCK and RT-qPCR for 7 patient samples and 4 controls. For **c**, **e**, see "Methods" for details about normalized fluorescence calculations. For **b**, **d**, **f**, **g**, NTC non-template control. For **a**, **c**, center = mean for 2 technical replicates. For **d–f**, center = mean and error bars = s.d. for 3 technical replicates. For **b**, **d**, RNA input at $10^4$ cp/μL. For **a–e**, **g**, source data are provided as a Source data file.

of HUDSON from 30 to 10 min for universal viral transport medium (UTM) and viral transport media (VTM), both used for NP swab samples, and for saliva (Fig. 3b and Supplementary Fig. 6). With this faster HUDSON protocol, we detected 50 cp/μL of synthetic RNA when spiked into UTM and 100 cp/μL when spiked into saliva, using a colorimetric readout (Supplementary Fig. 7). However, the lateral flow readout requires opening of tubes containing amplified products, which introduce risks of sample contamination. Thus, we incorporated an in-tube fluorescent readout with SHINE. Within 1 h, we detected as few as 10 cp/μL of SARS-CoV-2 synthetic RNA in HUDSON-treated UTM, 5 cp/μL in HUDSON-treated VTM, and 5 cp/μL in HUDSON-treated saliva with the in-tube fluorescent readout (Fig. 3c, d and Supplementary Figs. 8 and 9). To reduce user bias in interpreting results of this in-tube readout, we developed a companion smartphone app that uses the built-in smartphone camera to image the illuminated reaction tubes. The application then calculates the distance of the experimental tube's pixel intensity distribution from that of a user-selected negative control tube and returns a binary result indicating the presence or absence of viral

RNA in the sample (Fig. 3a, e; see "Methods" for details). Thus SHINE both minimizes equipment requirements and user interpretation bias when implemented with this in-tube readout and the smartphone application.

**Assessment of SHINE's performance on patient samples**. We used SHINE to test a set of 50 unextracted NP samples from 30 SARS-CoV-2-positive patients with samples previously tested and confirmed by RT-qPCR and 20 SARS-CoV-2-negative patients. First, we used SHINE with the paper-based colorimetric readout on a subset of 6 SARS-CoV-2-positive samples and detected SARS-CoV-2 RNA in all 6 positive samples, and in none of the negative controls (100% concordance, Fig. 3f). Subsequently, for all 50 samples, we used SHINE with the in-tube fluorescence readout and companion smartphone application. We detected SARS-CoV-2 RNA in 27 of the 30 COVID-19-positive samples (90% sensitivity) and none of the COVID-19-negative samples (100% specificity) after a 10-min HUDSON and a 40-min single-step SHERLOCK incubation (Fig. 3g, h, Supplementary Fig. 10,

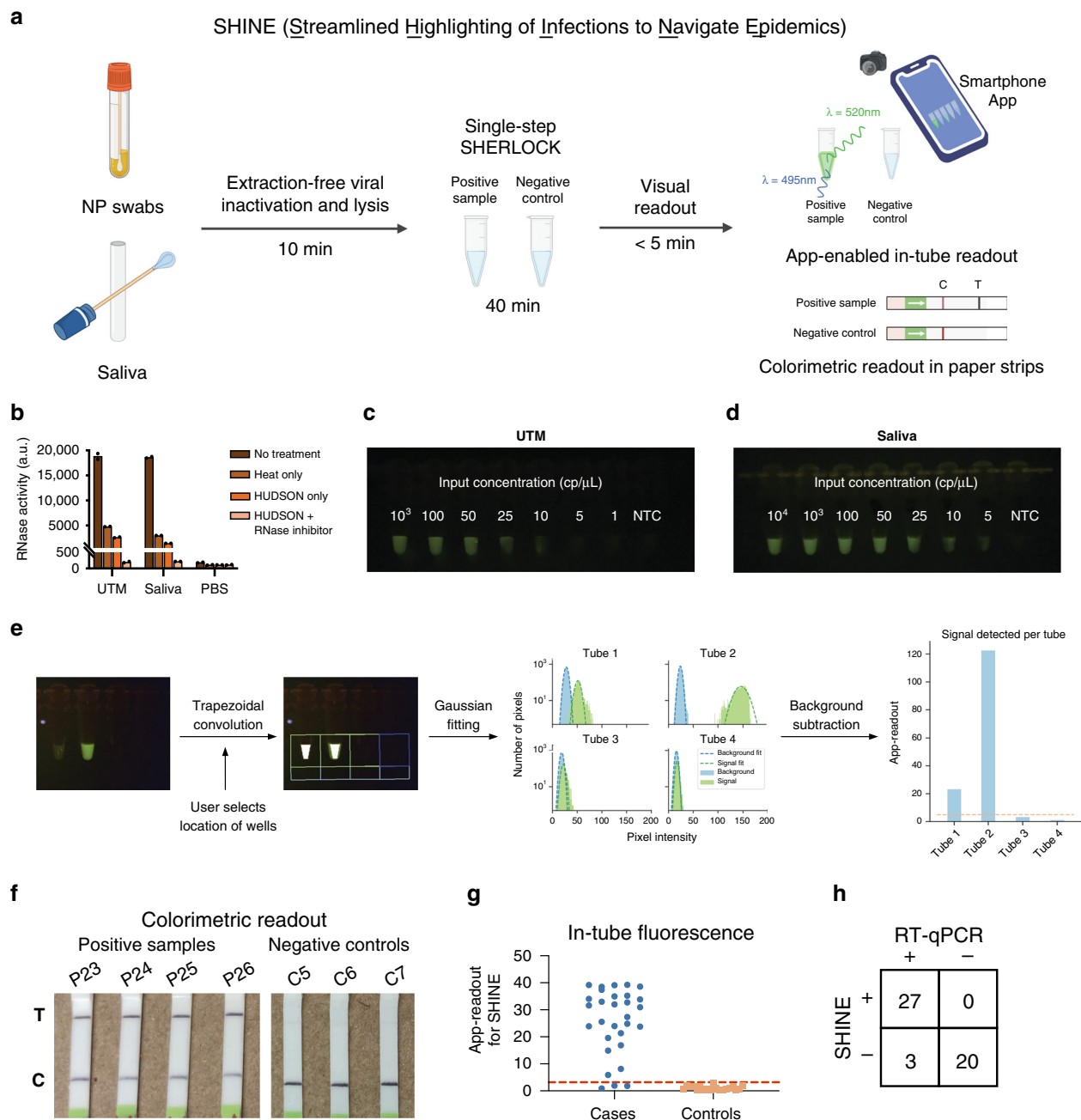

**Fig. 3 SARS-CoV-2 detection from unextracted samples using SHINE. a** Schematic of SHINE, which streamlines SARS-CoV-2 detection by using HUDSON to inactivate samples and single-step SHERLOCK to detect viral RNA with an in-tube fluorescent or colorimetric readout. Times suggested incubation times, C control line, T test line. **b** Measurement of RNase activity using RNaseAlert after 30 min at room temperature from treated or untreated universal viral transport medium (UTM), saliva, and phosphate-buffered saline (PBS). **c** SARS-CoV-2 RNA detection in UTM using SHINE with the in-tube fluorescence readout after 1 h. **d** SARS-CoV-2 RNA detection in saliva using SHINE with the in-tube fluorescence readout after 1 h. **e** Schematic of the companion smartphone application for quantitatively analyzing in-tube fluorescence and reporting binary outcomes of SARS-CoV-2 detection. **f** Colorimetric detection of SARS-CoV-2 RNA in unextracted patient NP swabs using SHINE after 1 h. **g** SARS-CoV-2 detection from 50 unextracted patient samples using SHINE and smartphone application quantification of in-tube fluorescence after 40 min. Threshold line plotted as mean readout value for controls plus 3 standard deviations. **h** Concordance table between SHINE and RT-qPCR for 50 patient samples. For **b**, center = mean for 2 technical replicates. For **b**, **g**, source data are provided as a Source data file.

and Supplementary Tables 1 and 2). Thus SHINE demonstrated 94% concordance using the in-tube readout with a total run time of 50 min. Notably, the RT-qPCR-positive patient NP swabs that SHINE failed to detect have higher Ct values than those that SHINE detected as positive ($p = 0.0017$ via one-sided Wilcoxon rank-sum test; Supplementary Fig. 11).

To assess our LOD and assay variability across replicates with patient samples, we tested SHINE on a set of 12 independent, unextracted NP samples of varying viral titer as determined by RT-qPCR. For these 12 samples, we performed SHINE and the Centers for Disease Control and Prevention (CDC) RT-qPCR N1 assay[30] on identical sample aliquots to eliminate potential differences in titer

due to uneven numbers of freeze–thaw cycles. We found that samples with titers of <100 cp/μL were not detected (4 of the 12 samples), those with titers ranging from 100 to 1000 cp/μL were detected in one or more technical replicates (4 of the 12 samples), and those with titers >1000 cp/μL were detected in all technical replicates (4 of the 12 samples) (Supplemental Fig. 12). Therefore, SHINE's performance was equivalent to the CDC assay for RT-qPCR-imputed titers >1000 cp/μL, but multiple replicates are needed for samples with lower titers. Importantly, SHINE's sensitivity on patient samples falls well within the range suggested for screening in reopening settings, while offering the rapid turnaround time necessary for testing at a frequency as high as daily[31].

## Discussion

Here we describe SHINE, a simple method for detecting viral RNA from unextracted patient samples with minimal equipment requirements and multiple readouts. SHINE's simplicity matches that of the most streamlined nucleic acid diagnostics while other isothermal methods require nucleic acid purification or additional readout steps. The use of HUDSON for both NP swabs and less invasive sample types, like saliva, greatly simplifies sample processing. Furthermore, SHINE's performance with saliva is particularly important as it reliably contains SARS-CoV-2 RNA and is ideal for routine or daily testing[32–34]. SHINE's two readouts, lateral flow and in-tube fluorescence, have tradeoffs between equipment needs and sample batch size. Specifically, the lateral flow readout reduces equipment requirements to solely a heat block, but requires longer incubation times to detect samples with lower viral titers. This readout is less amenable to testing large numbers of samples simultaneously and introduces potential risk of sample cross-contamination, as lateral flow strips must be manually inserted into an opened tube for each sample. In contrast, many samples can be imaged in parallel using the in-tube fluorescence readout, but a blue light-emitting device is required. The use of portable transilluminators (0.45 kg in weight for < $500) or small, blue LED lights (~$15) would eliminate the need for large or expensive fluorescent readers[35]. Furthermore, the in-tube fluorescence readout and companion smartphone application lend themselves to automated interpretation of results, which is both unbiased and fast. We believe that SHINE is particularly well suited for community surveillance testing, as it combines user-friendly, simple preparation methods with sufficient sensitivity and a rapid turn-around time.

With the improvements described, CRISPR-based assays have the potential to address diagnostic needs during the COVID-19 pandemic and in outbreaks to come. Previously developed CRISPR-based detection methods for COVID-19 are highly sensitive and specific, but these assays were primarily tested with purified nucleic acid and require multiple sample-manipulation steps[19,20,24,26,28,36,37]. SHINE addresses these limitations, requiring solely two reaction mixtures and sample transfer steps for sample processing and viral detection. With SHINE, CRISPR-based diagnostic testing can now be high-throughput while still only requiring portable equipment, highlighting the technology's potential to disrupt the centralized testing model for diagnosis of infections.

Comparing the performance of SHINE to the gold-standard RT-qPCR methods is essential for understanding its utility for clinical testing. Notably, SHINE demonstrates perfect concordance with RT-qPCR in our samples with titers >1000 viral cp/μL. However, it does exhibit stochasticity both across and within the lower-titer samples. The association of RT-qPCR Ct value with SHINE's performance suggests that some of the observed non-concordance in test results may be due to assay sensitivity or

degradation of sample material associated with an additional freeze–thaw cycle, as the two assays were not performed side-by-side for the majority of patient samples. Non-concordance could also be due to differences in sample processing or assay design. The SHINE and RT-qPCR assays are designed to detect different SARS-CoV-2 open reading frames (ORF), and Ct values for both genes are unlikely to be equivalent. Furthermore, metagenomic sequencing of COVID-19-postive patient samples has revealed that many samples with higher Ct values do not result in full coverage across the genome[38]. Variation in the levels of each ORF or genomic region as well as differences in upstream sample processing may explain the observed differences between the determined LOD using synthetic RNA targets compared to that of patient samples. Improved inactivation methods that allow for increased sample volume input or additional modifications to the single-step SHERLOCK could make these methods more comparable in performance.

Additional advances are still required for highly sensitive diagnostic testing to occur in virtually any location with a rapid turnaround time. Ideally, all steps would be performed at ambient temperature in ≤15 min and via a colorimetric readout that does not require tube opening. Existing nucleic acid diagnostics, to our knowledge, are not capable of meeting all these requirements simultaneously. Sample collection without UTM (i.e., "dry swabs") combined with spin-column-free extraction buffers and incorporation of solution-based, colorimetric readouts could address these limitations[37,39–41]. Solution-based visual readouts are additionally valuable because of reduced risk of contamination across samples containing amplified products. Ultimately, formulations of SHINE would be lyophilized, which would simplify distribution and assay preparation, and allow tests to be shelf-stable. Together, these advances could greatly enhance the accessibility of diagnostic testing and provide an essential tool in the fight against infectious diseases. By reducing personnel time, equipment, and assay time to results without sacrificing sensitivity or specificity, we have taken steps toward the development of such a tool.

## Methods

**Reagents and materials.** Detailed information about reagents, including the commercial vendors and stock concentrations, is provided in Supplementary Table 3.

**Clinical samples and ethics statement.** Clinical samples were de-identified and acquired from clinical studies evaluated and approved by the Institutional Review Board/Ethics Review Committee of the Massachusetts General Hospital and Massachusetts Institute of Technology (MIT) or Redeemer's University Ethical Review Committee. De-identified clinical samples from Boca Biolistics were obtained commercially under their ethical approvals. The Office of Research Subject Protection at the Broad Institute of MIT and Harvard University approved the use of samples included in this study.

**Viral and extracted sample preparation and RT-qPCR testing.** For side-by-side comparisons of the two-step SHERLOCK assay and RT-qPCR on viral seedstocks, the 2019-nCoV/USA-WA1-A12/2020 isolate of SARS-CoV-2 was provided by the US CDC. The virus was passaged at the Integrated Research Facility-Frederick in high containment (BSL-3) by inoculating grivet kidney epithelial Vero cells (American Type Culture Collection (ATCC) #CCL-81) at a multiplicity of infection of 0.01. Infected cells were incubated for 48 or 72 h in Dulbecco's Modified Eagle Medium with 4.5 g/L D-glucose, L-glutamine, and 110 mg/L sodium pyruvate (Gibco) containing 2% heat-inactivated fetal bovine serum (SAFC Biosciences) in a humidified atmosphere at 37 °C with 5% $CO_2$. The resulting viral stock was harvested and quantified by plaque assay using Vero E6 cells (ATCC #CRL-1586) with a 2.5% Avicel overlay and stained after 48 h with a 0.2% crystal violet stain.

For side-by-side comparisons of the two-step SHERLOCK assay and RT-qPCR on patient samples, nasal swab or combined nasal and saliva samples were collected from symptomatic patients in whom COVID-19 was suspected. Nasal swabs were collected and stored in viral transport medium (VTM)[42]. All nucleic acid extractions were performed using the QIAamp Viral RNA Mini Kit (Qiagen). For a subset of patients, saliva samples were combined with nasal samples during extraction. The starting volume for extraction was 70 μL and extracted nucleic acid

was eluted into 60 µL of nuclease-free water. RT-qPCR was performed using either the RT-PCR Reagent Set for COVID-19 Real-time detection (DaAn-GENE) or the GeneFinder™ COVID-19 Plus RealAmp Kit (OSANG Healthcare) using the N target (primer and probe sequences not publically available). RT-qPCR cycling conditions for the DnAn-GENE Kit were as follows: RT at 50 °C for 15 min, heat activation at 95 °C for 15 min and 45 cycles of a denaturing step at 94 °C for 15 s followed by annealing and elongation steps at 55 °C for 45 s. RT-qPCR cycling conditions for the OSANG Healthcare's Kit were as follows: RT at 50 °C for 20 min, heat activation at 95 °C for 5 min, and 45 cycles with a denaturing step at 95 °C for 15 s followed by annealing and elongation steps at 58 °C for 60 s.

Nasal swabs were collected and stored in UTM (BD) or VTM and stored at −80 °C prior to nucleic acid extraction. For the initial set of 50 NP patient samples, nucleic acid extraction was performed using MagMAX™ mirVana™ Total RNA Isolation Kit. The starting volume for the extraction was 250 µL and extracted nucleic acid was eluted into 60 µL of nuclease-free water. Extracted nucleic acid was then immediately Turbo DNase-treated (Thermo Fisher Scientific), purified twice with RNACleanXP SPRI beads (Beckman Coulter), and eluted in 15 µL of Ambion Linear Acrylamide (Thermo Fisher Scientific) water (0.8%).

Turbo DNase-treated extracted RNA was then tested for the presence of SARS-CoV-2 RNA using a laboratory-developed, probe-based RT-qPCR assay based on the N1 target of the CDC assay[30]. RT-qPCR was performed on a 1:3 dilution of the extracted RNA using TaqPath™ 1-Step RT-qPCR Master Mix (Thermo Fisher Scientific) with the following forward and reverse primer sequences, respectively: forward GACCCCAAAATCAGCGAAAT, reverse TCTGGTTACTGCCAGTTG AATCTG. The RT-PCR assay was run with a double-quenched FAM probe with the following sequence: 5′-FAM-ACCCCGCATTACGTTTGGTGGACC-BHQ1-3′. RT-qPCR was run on a QuantStudio 6 (Applied Biosystems) with RT at 48 °C for 30 min and 45 cycles with a denaturing step at 95 °C for 10 s followed by annealing and elongation steps at 60 °C for 45 s. The data were analyzed using the Standard Curve module of the Applied Biosystems Analysis Software.

Patient samples for side-by-side SHINE and RT-qPCR testing (from Boca Biolistics) were extracted using the QIAamp Viral RNA Mini Kit (Qiagen). The starting volume for the extraction was 100 µL and extracted nucleic acid was eluted into 40 µL of nuclease-free water. Extracted RNA was then tested for the presence of SARS-CoV-2 RNA using the laboratory-developed, probe-based RT-qPCR assay mentioned above (based on the N1 target of the CDC assay). Primers, probes, and conditions are the same as mentioned above.

**SARS-CoV-2 assay design and synthetic template information.** SARS-CoV-2-specific forward and reverse RPA primers and Cas13-crRNAs were designed as previously described[19]. In short, the designs were algorithmically selected, targeting 100% of all 20 publicly available SARS-CoV-2 genomes at the time, and predicted by a machine learning model to be highly active (Metsky et al., in preparation). Moreover, the crRNA was selected for its high predicted specificity toward detection of SARS-CoV-2, versus related viruses, including other bat and mammalian coronaviruses and other human respiratory viruses (https://adapt.sabetilab.org/covid-19/).

Specificity target sequences were generated using the same design software noted above by providing the amplicon coordinates of the designed assay within the viral species of interest and an alignment of the selected viral species. The specificity targets tested represent the overall medoid of sequence clusters at the provided amplicon for each selected viral species within the designed SARS-CoV-2 SHERLOCK assay.

Synthetic DNA targets with appended upstream T7 promoter sequences (5′-GAAATTAATACGACTCACTATAGGG-3′) were ordered as double-stranded DNA (dsDNA) gene fragments from IDT and were in vitro transcribed to generate synthetic RNA targets. In vitro transcription was conducted using the HiScribe T7 High Yield RNA Synthesis Kit (New England Biolabs (NEB)) as previously described[24]. In brief, a T7 promoter ssDNA primer (5′-GAAATTAATACGACT CACTATAGGG-3′) was annealed to the dsDNA template and the template was transcribed at 37 °C overnight. Transcribed RNA was then treated with RNase-free DNase I (QIAGEN) to remove any remaining DNA according to the manufacturer's instructions. Finally, purification occurred using RNAClean SPRI XP beads at 2× transcript volumes in 37.5% isopropanol.

Sequence information for the synthetic targets, RPA primers, and Cas13-crRNA is listed in Supplementary Table 4.

**Two-step SARS-CoV-2 assay.** The two-step SHERLOCK assay was performed as previously described[19,24,26]. Briefly, the assay was performed in two steps: (1) isothermal amplification via RPA and (2) LwaCas13a-based detection using a single-stranded RNA (ssRNA) fluorescent reporter. For RPA, the TwistAmp Basic Kit (TwistDx) was used as previously described (i.e., with RPA forward and reverse primer concentrations of 400 nM and a magnesium acetate concentration of 14 mM)[26] with the following modifications: RevertAid reverse transcriptase (Thermo Fisher Scientific) and murine RNase inhibitor (NEB) were added at final concentrations of 4 U/µL each, and synthetic RNAs or viral seedstocks were added at known input concentrations making up 10% of the total reaction volume. The RPA reaction was then incubated on the thermocycler for 20 min at 41 °C. For the detection step, 1 µL of RPA product was added to 19 µL detection master mix. The detection master mix consisted of the following reagents (final concentrations

in master mix listed), with magnesium chloride added last: 45 nM LwaCas13a protein resuspended in 1× storage buffer (SB: 50 mM Tris pH 7.5, 600 mM NaCl, 5% glycerol, and 2 mM dithiothreitol (DTT); such that the resuspended protein is at 473.7 nM), 22.5 nM crRNA, 125 nM RNaseAlert substrate v2 (Thermo Fisher Scientific), 1× cleavage buffer (CB: 400 mM Tris pH 7.5 and 10 mM DTT), 2 U/µL murine RNase inhibitor (NEB), 1.5 U/µL NextGen T7 RNA polymerase (Lucigen), 1 mM of each rNTP (NEB), and 9 mM magnesium chloride. Reporter fluorescence kinetics were measured at 37 °C on a Biotek Cytation 5 plate reader using a monochromator (excitation: 485 nm, emission: 520 nm) every 5 min for up to 3 h.

**Single-step SARS-CoV-2 assay optimization.** The starting point for optimization of the single-step SHERLOCK assay was generated by combining the essential reaction components of both the RPA and the detection steps in the two-step assay, described above[24,26]. Briefly, a master mix was created with final concentrations of 1× original reaction buffer (20 mM HEPES pH 6.8 with 60 mM NaCl, 5% PEG, and 5 µM DTT), 45 nM LwaCas13a protein resuspended in 1× SB (such that the resuspended protein is at 2.26 µM), 136 nM RNaseAlert substrate v2, 1 U/µL murine RNase inhibitor, 2 mM of each rNTP, 1 U/µL NextGen T7 RNA polymerase, 4 U/µL RevertAid reverse transcriptase, 0.32 µM forward and reverse RPA primers, and 22.5 nM crRNA. The TwistAmp Basic Kit lyophilized reaction components (1 lyophilized pellet per 102 µL final master mix volume) were resuspended using the master mix. After pellet resuspension, cofactors magnesium chloride and magnesium acetate were added at final concentrations of 5 and 17 mM, respectively, to complete the reaction.

Master mix and synthetic RNA template were mixed and aliquoted into a 384-well plate in triplicate, with 20 µL per replicate at a ratio of 19:1 master mix:sample. Fluorescence kinetics were measured at 37 °C on a Biotek Cytation 5 or Biotek Synergy H1 plate reader every 5 min for 3 h, as described above. We observed no significant difference in performance between the two plate reader models.

Optimization occurred iteratively, with a single reagent modified in each experiment. The reagent condition (e.g., concentration, vendor, or sequence) that produced the most optimal results—defined as either a lower LOD or improved reaction kinetics (i.e., reaction saturates faster)—was incorporated into our protocol. Thus the protocol used for every future reagent optimization consisted of the most optimal reagent conditions for every reagent tested previously.

For all optimization experiments, the modulated reaction component is described in the figures, associated captions, or associated legends. Across all experiments, the following components of the master mix were held constant: 45 nM LwaCas13a protein resuspended in 1× SB (such that the resuspended protein is at 2.26 µM), 1 U/µL murine RNase inhibitor, 2 mM of each rNTP, 1 U/µL NextGen T7 RNA polymerase, and 22.5 nM crRNA, and TwistDx RPA TwistAmp Basic Kit lyophilized reaction components (1 lyophilized pellet per 102 µL final master mix volume). In all experiments, the master mix components except for the magnesium cofactor(s) were used to resuspend the lyophilized reaction components, and the magnesium cofactor(s) were added last. All other experimental conditions, which differ among the experiments due to real-time optimization, are detailed in Supplementary Table 5.

**Single-step SARS-CoV-2 optimized reaction.** The optimized reaction (see Supplementary Protocol for exemplary implementation) consists of a master mix with final concentrations of 1× optimized reaction buffer (20 mM HEPES pH 8.0 with 60 mM KCl and 5% PEG), 45 nM LwaCas13a protein resuspended in 1× SB (such that the resuspended protein is at 2.26 µM), 125 nM polyU [i.e., 6 uracils (6U) or 7 uracils (7U) in length, unless otherwise stated] FAM quenched reporter, 1 U/µL murine RNase inhibitor, 2 mM of each rNTP, 1 U/µL NextGen T7 RNA polymerase, 2 U/µL Invitrogen SuperScript IV (SSIV) reverse transcriptase (Thermo Fisher Scientific), 0.1 U/µL RNase H (NEB), 120 nM forward and reverse RPA primers, and 22.5 nM crRNA. Once the master mix is created, it is used to resuspend the TwistAmp Basic Kit lyophilized reaction components (1 lyophilized pellet per 102 µL final master mix volume). Finally, magnesium acetate is the sole magnesium cofactor and is added at a final concentration of 14 mM to generate the final master mix.

The sample is added to the complete master mix at a ratio of 1:19, and the fluorescence kinetics are measured at 37 °C using a Biotek Cytation 5 or Biotek Synergy H1 plate reader as described above.

For the specificity data, fluorescence kinetics were measured at 37 °C using a Molecular Devices SpectraMax M2 plate reader using the same excitation and emission parameters described above; notably, this plate reader model required twice the reporter concentration (250 nM polyU FAM) to achieve a comparable LOD to the Biotek models.

**Detection via in-tube fluorescence and lateral flow strip.** Minor modifications were made to the optimized single-step and the two-step SARS-CoV-2 reaction to visualize the readout via in-tube fluorescence or lateral flow strip.

For in-tube fluorescence with the optimized single-step reaction, we generated the master mix as described above, except the 7U FAM quenched reporter was used at a concentration of 62.5 nM. The sample was added to the complete master mix at a ratio of 1:19. Samples were incubated at 37 °C, and images were collected after 30, 45, 60, 90, 120 and 180 min of incubation, with image collection terminating

once experimental results were clear. A dark reader transilluminator (DR196 model, Clare Chemical Research) or Gel Doc™ EZ Imager (BioRad) with the blue tray was used to illuminate the tubes.

For lateral flow readout with the two-step SHERLOCK method, we generated the Cas13-based detection mix as described above, except we used a biotinylated FAM reporter at a final concentration of 1 μM rather than RNase Alert v2. For lateral flow readout using the optimized single-step SHERLOCK assay, we generated the single-step master mix as described above, except we used a biotinylated FAM reporter at a final concentration of 1 μM rather than the quenched polyU FAM reporters. For both two-step and single-step SHERLOCK, the sample was added to the complete master mix at a ratio of 1:19. After 1–3 h of incubation at 37 °C, the detection reaction was diluted 1:4 in Milenia HybriDetect Assay Buffer, and the Milenia HybriDetect 1 (TwistDx) lateral flow strip was added. Sample images were collected 5 min following incubation of the strip. Lateral flow results were assessed either by the user or in an automated fashion by measuring the pixel intensity of the test band using ImageJ.

**In-tube fluorescence reader mobile phone application**. To enable smartphone-based fluorescence analysis, we designed a companion mobile application pipeline. Using the application, the user captures an image of a set of strip tubes illuminated by a transilluminator. The user then identifies regions of interest in the captured image by overlaying a set of pre-drawn boxes onto experimental and control tubes. Image and sample information is then transmitted to a server for analysis. Within each of the user-selected squares, the server models the bottom of each tube as a trapezoid and uses a convolutional kernel to determine the location of maximal signal within each tube, using data from the green channel of the RGB image. The server then identifies the background signal proximal to each tube and fits a Gaussian distribution around the background signal and around the in-tube signal. The difference between the mean pixel intensity of the background signal and the mean pixel intensity of the in-tube signal is then calculated as the background-subtracted fluorescence signal for each tube. To identify experimentally significant fluorescent signals, a score is computed for each experimental tube; this score is equal to the distance between the experimental and control background-subtracted fluorescence divided by the standard deviation of pixel intensities in the control signal. Finally, positive or negative samples are determined based on whether the score is above (positive, +) or below (negative, −) 1.5, a threshold identified empirically.

**HUDSON protocols**. HUDSON nuclease and viral inactivation were performed on viral seedstock as previously described with minor modifications to the temperatures and incubation times[25]. In short, 100 mM TCEP (Thermo Fisher Scientific) and 1 mM EDTA (Thermo Fisher Scientific) were added to non-extracted viral seedstock and incubated for 20 min at 50 °C, followed by 10 min at 95 °C. The resulting product was then used as input into the two-step SHERLOCK assay.

The improved HUDSON nuclease and viral inactivation protocol was performed as previously described, with minor modifications[26]. Briefly, 100 mM TCEP, 1 mM EDTA, and 0.8 U/μL murine RNase inhibitor were added to clinical samples in UTM, VTM, or human saliva (Lee Biosolutions). These samples were incubated for 5 min at 40 °C, followed by 5 min at 70 °C (or 5 min at 95 °C, if saliva). The resulting product was used in the single-step detection assay. In cases where synthetic RNA targets were used, rather than clinical samples (e.g., during reaction optimization), targets were added after the initial heating step (40 °C at 5 min). This is meant to recapitulate patient samples, as RNA release occurs after the initial heating step when the temperature is increased and viral particles lyse.

For optimization of nuclease inactivation using HUDSON, only the initial heating step was performed. The products were then mixed 1:1 with 400 mM RNaseAlert substrate v2 in nuclease-free water and incubated at room temperature for 30 min before imaging on a transilluminator or measuring reporter fluorescence on a Biotek Synergy H1 [at room temperature using a monochromator (excitation: 485 nm, emission: 520 nm) every 5 min for up to 30 min]. The specific HUDSON protocol parameters modified are indicated in the figure captions.

**Data analysis and schematic generation**. Conservation of SARS-CoV-2 sequences across our SHERLOCK assay was determined using publicly available genome sequences via GISAID. Analysis was based on an alignment of 5376 SARS-CoV-2 genomic sequences. Percent conservation was measured at each nucleotide within the RPA primer and Cas13-crRNA-binding sites and represents the percentage of genomes that have the consensus base at each nucleotide position.

As described above, fluorescence values are reported as background-subtracted, with the fluorescence value collected before reaction progression (i.e., the latest time at which no change in fluorescence is observed, usually time 0, 5, or 10 min) subtracted from the final fluorescence value (3 h, unless otherwise indicated).

Normalized fluorescence values are calculated using data aggregated from multiple experiments with at least one condition in common and for the specificity testing where all conditions were performed in the same experiment on a SpectraMax M2 (Molecular Devices). The maximal fluorescence value across all experiments is set to 1, with fluorescence values from the same experiment set as ratios of the maximal fluorescence value. Common conditions across experiments

are set to the same normalized value, and that value is propagated to determine the normalized values within an experiment.

The Wilcoxon rank-sum test was conducted in MATLAB (MathWorks). Schematics shown in Figs. 1a and 3a were created using www.biorender.com. All other schematics were generated in Adobe Illustrator (v24.1.2). Data panels were primarily generated via Prism 8 (GraphPad), except Fig. 3e that was generated using Python (version 3.7.2), seaborn (version 0.10.1), and matplotlib (version 3.2.1)[43,44].

**Reporting summary**. Further information on research design is available in the Nature Research Reporting Summary linked to this article.

## Data availability

The data, code, and detailed methods used in the design of primers and crRNAs are available at adapt.sabetilab.org. Any other relevant data are available from the authors upon reasonable request. Source data are provided with this paper.

## Code availability

The code for the smartphone application analysis pipeline is available at https://github.com/broadinstitute/Handlens.

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

## Acknowledgements

We would like to thank E. Rosenberg for kindly providing patient samples used in this study; the Harvard Medical School Systems Biology Department, Harvard University Northwest Labs, and A. Viel for providing additional laboratory space to perform the work; those researchers and laboratories who generously made SARS-CoV-2 sequencing data publicly available to aid in our assay design; members of the Sabeti laboratory—N. Welch, E. Normandin, K. DeRuff, K. Lagerborg, M. Bauer, M. Rudy, K. Siddle, A. Lin, and A. Gladden-Young—for assisting with patient sample collection and processing; H. Metsky, for his contributions to the assay design; M. Springer, the Springer laboratory, and the Sabeti laboratory, notably H. Metsky, A. Lin, and N. Welch for their thoughtful discussions and reading of the manuscript. Funding was provided by DARPA D18AC00006 and the Open Philanthropy Project. J.A.-S. is supported by a fellowship from "la Caixa" Foundation (ID 100010434, code LCF/BQ/AA18/11680098). B.A.P. is supported by the National Institute of General Medical Sciences grant T32GM007753. For L.W., K.C., and N.H. B., this work was funded under Agreement No. HSHQDC-15-C-00064 awarded to Battelle National Biodefense Institute (BNBI) by the Department of Homeland Security (DHS) Science and Technology (S&T) Directorate for the management and operation of the National Biodefense Analysis and Countermeasures Center (NBACC), a Federally Funded Research and Development Center. The views, opinions, conclusions, and/or findings expressed should not be interpreted as representing the official views or policies, either expressed or implied of the Department of Defense, US government, National Institute of General Medical Sciences, DHS, or the National Institutes of Health. The DHS does not endorse any products or commercial services mentioned in this presentation. In no event shall the DHS, BNBI or NBACC have any responsibility or liability for any use, misuse, inability to use, or reliance upon the information contained herein. In addition, no warranty of fitness for a particular purpose, merchantability, accuracy or adequacy is provided regarding the contents of this document.Notice: This manuscript has been authored by Battelle National Biodefense Institute, LLC under Contract No. HSHQDC-15-C-00064 with the U.S. Department of Homeland Security. The United States Government retains and the publisher, by accepting the article for publication, acknowledges that the United States Government retains a non-exclusive, paid up, irrevocable, world-wide license to publish or reproduce the published form of this manuscript, or allow others to do so, for United States Government purposes.

## Author contributions

J.A.-S. and C.A.F. conceived the study under the guidance and supervision of P.C.S. and C.M. J.A.-S. and G.A. performed experiments and data analysis for preliminary single-step experiments. T.-S.F.K.-T., J.N.U., F.V.A., P.E.E., R.B., L.W., and K.C. performed experiments and data analysis for initial two-step SARS-CoV-2 SHERLOCK experiments. J.A.-S., C.A.F., A.C.S., C.K.B., and B.A.P. designed and performed optimization for single-step SARS-CoV-2 SHERLOCK and completed data analysis. J.A.-S., A.C.S., C.K.B., and B.A.P. performed HUDSON and SHINE experiments and completed data analysis. S.S. developed the smartphone application. J.A.-S., B.A.P., J.N.U., and F.V.A. performed and analyzed experiments with patient samples. B.M.S., G.A., P.E.E., and J.E.L. provided assistance in patient sample collection. M.E.K., L.E.H., N.H.B., B.L.M., and C.T.H. provided critical insights on protocols, the results, and the work. J.A.-S., C.A.F., and B.A.P. wrote the paper with guidance from P.C.S. and C.M. All authors reviewed the manuscript. Correspondence can also be sent to C.M. at cmyhrvol@princeton.edu.

## Competing interests

C.A.F., P.C.S., and C.M. are inventors on patent PCT/US2018/022764, which covers the SHERLOCK and HUDSON technology for viral RNA detection held by the Broad Institute. J.A.-S., C.A.F., A.C.S., B.A.P., P.C.S., and C.M. are inventors on a pending patent application held by the Broad Institute (U.S. Provisional Patent Application No. 63/074,307). This pending application covers the SHINE technology and all designed sequences used in this work. J.E.L. consults for Sherlock Biosciences, Inc. P.C.S. is a co-founder of, shareholder in, and advisor to Sherlock Biosciences, Inc., as well as a Board member of and shareholder in Danaher Corporation. All other authors declare no competing interests.
