## [Peer Review File · Nature Communications]

Reviewers' Comments:

Reviewer #1:

Remarks to the Author:

Arizti-Sanz and colleagues present a new diagnostic tool for the rapid detection of SARS-CoV-2 RNA from unextracted samples

(SHINE, Streamlined Highlighting of Infections to Navigate Epidemics). The identified optimal conditions to allow RPA-based amplification and Cas-13 based detection in a single step (SHERLOCK-based SARS CoV-2 assay). Furthermore, they improved HUDSON to rapidly inactivate viruses in nasopharyngeal swabs and saliva and they demonstrate that SHINE can detect SARS-CoV-2 RNA in these samples within 50 min with a paper-based colorimetric and an in-tube fluorescent readout using a companion smartphone application. They validated SHINE in 50 patient samples (30 SARS-coV-2 RT-PCR positive and 20 SARS-CoV-2 RT-PCR negative patients) and found a true positivity rate of 90% and a false positivity rate of 0%. Finally, they assessed the limit of detection and found that SHINE's performance using the in-tube fluorescent readout was equivalent to the CDC assay for RT-qPCR-imputed titers above 1000 copies/ μ L.

The findings of this manuscript are novel and of great interest for the community. The manuscript is very well written, methods are described in detail, and the results and discussion give a very good impression of the great potential of SHINE.

I only have two minor comments:

1) Your assay detected as few as 10 copies/ μ L when using the fluorescent readout and 100 copies/ μ L when using the lateral flow based readout (line 141-143). Why is the readout with the in-tube fluorescent readout with a detection limit of 1000 copies/ μ L in the patient samples higher (line 199)? When compared to RT-PCR assays a detection limit of \leq 100 copies/ μ L in patient samples would be great.

2) What are indications were you suggest to use SHINE? Just for screening or also for suspected COVID-19 cases?

Reviewer #2:

Remarks to the Author:

Arizti-Sanz et al. describe the development of a streamlined methodology for the Cas13-based detection of SARS-CoV-2. The current global pandemic shows a need for fast, reliable and point-of-care diagnostic solutions. The authors have taken the SHERLOCK assay and have adapted it for the detection of SARS-CoV-2 RNA. Firstly, they have designed and tested the so-called two-step assay which shows promising results on synthetic RNA, with a very high sensitivity of 10 copies/ μ L, which is comparable to that obtained by the 'gold standard' RT-qPCR method. Next, the authors describe attempts to generate a single-step assay based on SHERLOCK and on the sample extraction by the previously established HUDSON method. Combining the amplification step RT-RPA and the Cas13-based detection step in a single reaction, initially results in a single-step assay with a poor sensitivity of 6 orders of a magnitude lower (10^6 copies/ μ L) lower than the original 2-step approach. Next they set out to optimize the system by systematically varying a range of parameters in the reaction mixture. In several steps the authors succeeded in increasing the sensitivity around 100,000 fold, bringing it on par with the previously tested two-step assay. Further developments have been made by the authors on the previously described HUDSON protocol. The combined new methodology is called SHINE. On top of the SHINE protocol, the authors have developed a mobile-phone application that allows readout of fluorescent signals using the phone's camera. Final validation of SHINE has been done by testing the performance on patient samples, showing high concordance compared to the RT-qPCR control. As such, this work is a valuable contribution to the development of diagnostics for the global fight against the SARS-

CoV-2 outbreak.

Major concern

The manuscript is well-written and well presented. On top of that, the work presented is certainly timely. However, taking into account the high impact of the Nature Comm. journal, my major criticism is the lack of novelty in this manuscript. Despite the development of a very promising detection system, the main results are heavily based on combining previously developed systems, SHERLOCK (REF 23) and HUDSON (REF 25). In addition, the smartphone application has been described before (Nature Comm in press), the RPA optimization is described in a co-submitted manuscript, and Cas13 guide selection is described elsewhere (in prep). All in all, the step-wise optimization (adding RNaseH, varying buffer and primer concentration, using a poly-U reporter, and replacing the RT enzyme) results in a very impressive optimization of the system, but in my opinion this does not warrant publication in Nature Communications.

Minor concerns

Line 104: "but preparing the reactions required 45-90 minutes of hands-on time depending on the number of samples". What is this time based on, seems like a gross overestimation of required time. Please clarify.

Supplementary Fig 2 C

Please clarify both the Figure and the legend.

Fig 3b

In the current presentation of the Figure it appears that samples with no treatment perform the best. Higher fluorescence is best? Please clarify.

Reviewer #1

Arizti-Sanz and colleagues present a new diagnostic tool for the rapid detection of SARS-CoV-2 RNA from unextracted samples (SHINE, Streamlined Highlighting of Infections to Navigate Epidemics). The identified optimal conditions to allow RPA-based amplification and Cas-13 based detection in a single step (SHERLOCK-based SARS CoV-2 assay). Furthermore, they improved HUDSON to rapidly inactivate viruses in nasopharyngeal swabs and saliva and they demonstrate that SHINE can detect SARS-CoV-2 RNA in these samples within 50 min with a paper-based colorimetric and an in-tube fluorescent readout using a companion smartphone application. They validated SHINE in 50 patient samples (30 SARS-coV-2 RT-PCR positive and 20 SARS-CoV-2 RT-PCR negative patients) and found a true positivity rate of 90% and a false positivity rate of 0%. Finally, they assessed the limit of detection and found that SHINE's performance using the in-tube fluorescent readout was equivalent to the CDC assay for RT-qPCR-imputed titers above 1000 copies/ μ L.

The findings of this manuscript are novel and of great interest for the community. The manuscript is very well written, methods are described in detail, and the results and discussion give a very good impression of the great potential of SHINE.

We thank the reviewer for this wonderful summary of our manuscript, and the reviewer's positive comments about the novelty of this work and its interest to the community.

I only have two minor comments:

1) Your assay detected as few as 10 copies/ μ L when using the fluorescent readout and 100 copies/ μ L when using the lateral flow based readout (line 141-143). Why is the readout with the in-tube fluorescent readout with a detection limit of 1000 copies/ μ L in the patient samples higher (line 199)? When compared to RT-PCR assays a detection limit of \leq 100 copies/ μ L in patient samples would be great.

We thank the reviewer for raising this concern and have added text to the discussion to comment on the observed differences in the limit of detection between patient samples and our synthetic target testing (lines #251-258), reproduced here and elaborated on below:

Non-concordance could also be due to differences in sample processing or assay design. The SHINE and RT-qPCR assays are designed to detect different SARS-CoV-2 open-reading frames, and Ct values for both genes are unlikely to be equivalent. Furthermore, metagenomic sequencing of COVID-19-positive patient samples has revealed that many samples with higher Ct values do not result in full coverage across the genome³⁷. Variation in the levels of each open-reading frame or genomic region as well as differences in upstream sample processing may explain the observed differences between the determined limit of detection using synthetic RNA targets compared to that of patient samples.

Both sample processing methods and input volume amount differ between synthetic target and patient sample testing. Synthetic RNA targets were produced via *in vitro* transcription, quantified via a spectrophotometer, and directly added to sample matrices, yielding a precise measurement of copy number. However, the patient sample concentration was inferred via sample extraction, requiring purification and elution into a smaller volume, and subsequent RT-qPCR against the N target gene of SARS-CoV-2. Thus, the viral RNA concentration differs between the purified, RT-qPCR-quantified sample and the original sample, used as input for SHINE.

In summary, these factors contribute to SHINE's ability to detect SARS-CoV-2 in all replicates in patient samples with >1000 cp/μL N-gene RNA and in the majority of replicates in patient samples with >200 cp/μL.

2) What are indications were you suggest to use SHINE? Just for screening or also for suspected COVID-19 cases?

We thank the reviewer for this question and have added text in the discussion to comment on our suggested use cases for SHINE (lines #231-232).

Given the performance characteristics of our assay—in its current form—related to sensitivity, user ease, sample-to-answer time, and throughput, we suggest that this assay is best suited for surveillance applications. SHINE is easy to perform and can return results quickly, and therefore can be performed frequently. We also describe in lines #208-210 that SHINE's sensitivity on patient samples falls well within the range suggested for screening in reopening settings, while offering the rapid turnaround time necessary for testing at a frequency as high as daily (Paltiel *et al.* medRxiv, 2020). With further enhancements, it could be useful in the clinical setting.

Reviewer #2

Arizti-Sanz et al. describe the development of a streamlined methodology for the Cas13-based detection of SARS-CoV-2. The current global pandemic shows a need for fast, reliable and point-of-care diagnostic solutions. The authors have taken the SHERLOCK assay and have adapted it for the detection of SARS-CoV-2 RNA. Firstly, they have designed and tested the so-called two-step assay which shows promising results on synthetic RNA, with a very high sensitivity of 10 copies/uL, which is comparable to that obtained by the 'gold standard' RT-qPCR method. Next, the authors describe attempts to generate a single-step assay based on SHERLOCK and on the sample extraction by the previously established HUDSON method. Combining the amplification step RT-RPA and the Cas13-based detection step in a single reaction, initially results in a single-step assay with a poor sensitivity of 6 orders of a magnitude lower (10^6 copies/uL) lower than the original 2-step approach. Next they set out to optimize the system by systematically varying a range of parameters in the reaction mixture. In several steps the authors succeeded in increasing the sensitivity around 100,000 fold, bringing it on par with the previously tested two-step assay. Further developments have been made by the authors on the previously described HUDSON protocol. The combined new methodology is called SHINE. On top of the SHINE protocol, the authors have developed a mobile-phone application that allows readout of fluorescent signals using the phone's camera. Final validation of SHINE has been done by testing the performance on patient samples, showing high concordance compared to the RT-qPCR control. As such, this work is a valuable contribution to the development of diagnostics for the global fight against the SARS-CoV-2 outbreak.

We thank the reviewer for their comprehensive and thoughtful summary of our manuscript and their appreciation of the need for point-of-care diagnostics in the face of the SARS-CoV-2 outbreak.

Major concern

The manuscript is well-written and well presented. On top of that, the work presented is certainly timely. However, taking into account the high impact of the Nature Comm. journal, my major criticism is the lack of novelty in this manuscript. Despite the development of a very promising detection system, the main results are heavily based on combining previously developed systems, SHERLOCK (REF 23) and HUDSON (REF 25). In addition, the smartphone application has been described before (Nature Comm in press), the RPA optimization is described in a co-submitted manuscript, and Cas13 guide selection is described elsewhere (in prep). All in all, the step-wise optimization (adding RNaseH, varying buffer and primer concentration, using a poly-U reporter, and replacing the RT enzyme) results in a very impressive optimization of the system, but in my opinion this does not warrant publication in Nature Communications.

We thank the reviewer for their critical evaluation of the manuscript, but we believe our work presents a number of novel technological advances and critical results. Specifically:

1. This is the first published account of both isothermal amplification and CRISPR-based detection performed in a single reaction using RNA as input.
2. We substantially improve a rapid viral inactivation method that is compatible across multiple new sample types. This method is three times faster than the previously reported version of HUDSON.
3. We developed a new analysis pipeline for automated in-tube fluorescent result interpretation and expanded the smartphone application (app) that was developed for the lateral flow-based readout by our lab (Barnes *et al.*, *Nature Communications*) to include this new pipeline. Without this new analysis pipeline and expanded application, automated in-tube readout analysis is not possible. Notably, this analytic software interprets multiple samples in parallel and uses a statistical approach to compare sample tubes to control tubes. Integration of these two separate analysis pipelines adds functionality and allows researchers to interpret results using different readouts within a single app. To better distinguish this new software from that developed for lateral flow, we have restructured the end of the introduction in lines #87-90 and lines #96-99.
4. We performed extensive validation of SHINE against patient samples (48 SARS-CoV-2-positive samples, 20 SARS-CoV-2-negative samples).
5. We demonstrate the results from SARS-CoV-2 SHERLOCK detection from three different laboratories including one in Nigeria, highlighting the ability of this technology to be applied broadly.

Minor concerns

Line 104: “but preparing the reactions required 45-90 minutes of hands-on time depending on the number of samples”. What is this time based on, seems like a gross overestimation of required time. Please clarify.

We appreciate the reviewer’s comment and acknowledge that we used the term “hands-on time” to refer to what can be more comprehensively described as “reaction preparation time.”

To clarify this point, we have updated lines #110-112 of the text to the following:

Using both colorimetric and fluorescent readouts, we detected 10 cp/μL of synthetic RNA after incubating samples for 1 hour or less, but preparing the separate amplification and Cas13 detection reaction mixtures and combining each reaction mixture with each sample tested requires at least 45 minutes for a small number (<10) of samples (Fig. 1c, d and Supplementary Fig. 1a).

We also want to clarify how we determined this time range. We define reaction preparation time as the amount of time a user needs to set up the assay, including creating reaction mixtures and combining samples with reaction mixtures.

When determining the minimum time of 45 minutes, we assessed each step in the set up of two-step SHERLOCK reactions.

1. Preparing the RPA reaction mixtures requires minimally 15 minutes of set-up time.
2. Preparing the detection mixtures requires minimally 20 minutes as this reaction mixture has 10 components that are stored individually.
3. Combining each reaction mixture with appropriate sample volumes requires at least 5 minutes. These combinations are created at both the RPA and detection steps and require at least 10 minutes, with large batches of samples requiring more time.

Supplementary Fig 2 C

Please clarify both the Figure and the legend.

We have updated Supplementary Figure 2c to include a legend title of “RNase H concentration.”

We have also clarified the caption of Supplementary Figure 2c. It now reads as follows, with the added text underlined: “Background-subtracted fluorescence of the Cas13-detection reaction (no RPA) after 3 h incubation with varying RNase H concentrations.”

Fig 3b

In the current presentation of the Figure it appears that samples with no treatment perform the best. Higher fluorescence is best? Please clarify.

We apologize for this confusion. In Figure 3b, we are measuring the RNase activity of saliva or sample collection medium after different chemical and heat treatments to eliminate RNases. In this figure panel, higher fluorescence values indicate that RNases have not been sufficiently inactivated as the quenched RNA reporter (RNaseAlert), added following treatment, is cleaved by active RNases. Subsequently, lower fluorescence values are desired as these low values indicate that RNases have been sufficiently inactivated during HUDSON. See Methods, section “HUDSON protocols” (lines #516-521) for additional details on these experiments.

To eliminate this confusion, we have updated the axes of Figure 3b and associated figure caption reproduced below:

b

Measurement of RNase activity using RNaseAlert after 30 min at room temperature from treated or untreated universal viral transport medium (UTM), saliva, and phosphate buffered saline (PBS).

We have also added the following text in the results section at lines #163-165 to clarify the results and their interpretation: “During optimization, we assessed HUDSON’s ability to inactivate RNases by adding RNaseAlert to samples following treatment, with higher fluorescence corresponding to decreased nuclease inactivation.” We also updated the caption of Supplementary Figure 6 to clarify the results.